# Deriving household composition using population-scale electronic health record data—A reproducible methodology

**Rhodri D. Johnson** *, **Lucy J. Griffiths, Joe P. Hollinghurst, Ashley Akbari,**
**Alexandra Lee, Daniel A. Thompson, Ronan A. Lyons, Richard Fry**

Population Data Science, Swansea University Medical School, Swansea, United Kingdom

* r.d.johnson@swansea.ac.uk

## Abstract

### Background

Physical housing and household composition have an important role in the lives of individuals and drive health and social outcomes, and inequalities. Most methods to understand housing composition are based on survey or census data, and there is currently no reproducible methodology for creating population-level household composition measures using linked administrative data.

### Methods

Using existing, and more recent enhancements to the address-data linkage methods in the SAIL Databank using Residential Anonymised Linking Fields we linked individuals to properties using the anonymised Welsh Demographic Service data in the SAIL Databank. We defined households, household size, and household composition measures based on adult to child relationships, and age differences between residents to create relative age measures.

### Results

Two relative age-based algorithms were developed and returned similar results when applied to population and household-level data, describing household composition for 3.1 million individuals within 1.2 million households in Wales. Developed methods describe binary, and count level generational household composition measures.

### Conclusions

Improved residential anonymised linkage field methods in SAIL have led to improved property-level data linkage, allowing the design and application of household composition measures that assign individuals to shared residences and allow the description of household composition across Wales. The reproducible methods create longitudinal, household-level composition measures at a population-level using linked administrative data. Such measures are important to help understand more detail about an individual's home and area

**Data Availability Statement:** The data used for the study is third-party data and is held by the SAIL Databank at Swansea University on behalf of health care providers in Wales who are the original data

owners. This study was approved by the Secure Anonymised Information Linkage (SAIL) Information Governance Review Panel (project 1001) in Wales. All data were anonymised prior to access and analysis. We did not have special access to this data; it is available to anyone via an application to SAIL. All proposals to use SAIL data are subject to review by an independent Information Governance Review Panel (IGRP). Before any data can be accessed, approval must be given by the IGRP. The IGRP gives careful consideration to each project to ensure proper and appropriate use of SAIL data. When access has been approved, it is gained through a privacy-protecting safe haven and remote access system referred to as the SAIL Gateway. SAIL has established an application process to be followed by anyone who would like to access data via SAIL https://www.saildatabank.com/application-process.

**Funding:** This work was supported by the Medical Research Council (MRC) [MR/V028367/1]; Health Data Research (HDR) UK [HDR-9006] which receives its funding from the UK Medical Research Council, Engineering and Physical Sciences Research Council, Economic and Social Research Council, Department of Health and Social Care (England), Chief Scientist Office of the Scottish Government Health and Social Care Directorates, Health and Social Care Research and Development Division (Welsh Government), Public Health Agency (Northern Ireland), British Heart Foundation (BHF) and the Wellcome Trust; Administrative Data Research (ADR) UK which is funded by the Economic and Social Research Council [grant ES/S007393/1]; and the Nuffield Family Justice Observatory (FJO) [FJO/43766]. At the time when the work was undertaken, the authors were supported by these funders as follows: RDJ (ADR and Nuffield FJO); LJG (ADR UK, Nuffield FJO); JH (MRC); AA (HDR UK, ADR UK, MRC), AL (MRC, NFJO), RAL (HDR UK, ADR UK, MRC), RF (HDR UK, ADR UK, MRC). The funders had no role in study design, data collection and analysis, decision to publish, or preparation of the manuscript.

**Competing interests:** The authors have declared that no competing interests exist.

environment and how that may affect the health and wellbeing of the individual, other residents, and potentially into the wider community.

## Introduction

It is well understood that physical housing conditions and household composition play an important role in the health and wellbeing of individuals and are drivers of social and health inequalities [1]. Influences emanating from the home environment, whether residential or communal living, which may impact upon the individual are wide-ranging and include factors related to immediate and surrounding environments such as physical housing attributes, resident behaviours and characteristics, and household composition (HC). Household Composition Measures (HCM) describe the grouping of individuals in the same home, enabling household size and resident characteristics, including generational structures, to be calculated. The ability to quantify an individual's home environment and apply such HCM to individuals at a population-level has multiple and wide-ranging research applications.

HC has established impacts on: physical health in terms of contagious disease exposure and transmission patterns [2–6]; older age care, isolation, and role of family carers [7–10]; mental health [11–15]; social factors including deprivation [1], inequalities linked to overcrowding, multi-generational living and housing security [1, 11]; family justice and child wellbeing (effect on health, educational [16], maltreatment and child protection [17–22]); as well as providing further depth and granularity of the home environment for wider-ranging research.

Existing research using HCM are predominantly survey- or Census-based [3, 4], generally containing smaller numbers or sampled estimates, and are mainly cross-sectional at a fixed point in time, or use area-level distributions to estimate household structure [23]. HCM methods vary in detail from simple binary measures such as lone or multi households [18], household size combined with the presence of dependent children, to methods detailing residents' biological relationships, ethnicity, and other detailed household characteristics within inter-generational households [24, 25]. However, to the best of our knowledge, none have used routinely collected administrative data to ascertain a measure of HC or have the ability to use longitudinal data to understand changes in HC over time.

## Rationale

HCM need to be flexible to incorporate a variety of research outcomes. For instance, research aims may need to define households by allocating individuals to a property at the same time, calculate household size, or describe households through individual resident and household-level characteristics. In this paper, we describe enhancements to address linking methodology used in the Secure Anonymised Information Linkage (SAIL) Databank, and a methodology to model housing composition using the Welsh Demographic Service data at a population-level in Wales.

## Methods

This study was approved by the Secure Anonymised Information Linkage (SAIL) Information Governance Review Panel (project 1001) in Wales. All data were anonymised prior to access and analysis.

## Enhanced linking fields

The SAIL Databank is a privacy-protecting Trusted Research Environment (TRE) which houses routinely collected electronic health records (EHR) and administrative data sources relating to the population of Wales, United Kingdom (UK). However, it also comprises rich socio-economic and geographic data linked at individual-, and address-level. Linkage systems have been developed over time to incorporate new data and best practices. The Residential Anonymised Linking Field (RALF) was originally developed using the Royal Mail Postcode Address File (PAF) [26]. However, since 2013 RALF linkage improvements have been developed enabling more granular methods, both spatially and temporally. This has been achieved through the use of encrypted Unique Property Reference Numbers (UPRN) which are persistently unique through time and contain richer information on housing type, houses in multiple occupation, and communal residences. The enhanced RALF system also allows us to take advantage of the drive at UK Government level to make the UPRN the standard unique identifier for address-based data [27]. The methodology has allowed us to develop core data assets in SAIL which capture the different environments in which the people of Wales live and go about their daily lives. This has been achieved through a combination of administrative geospatial data attributes. These include:

- Care Home Anonymised Linking Fields (CH-ALF) allow grouping of people into discrete care homes and facilitates linking of further administrative data (e.g. size, number of places, services offered) and to health and social care data of residents.

- Workplace Anonymised Linking Field (W-ALF) allows grouping of people into discrete workplace settings and classification by industry type and setting.

- School Anonymised Linking Field (S-ALF) which allows pupils and teachers to be grouped in discrete educational settings.

Using such linking fields, sophisticated geospatial models of the built environment have been developed pertaining to a place of residence, education, or work which can be anonymously linked with a variety of data sources. A further benefit of the UPRN based linking fields is that, at a high level, it allows us to group individuals by discrete spatial units, namely their place of residence or RALF. However, the enhanced RALF can't model relationships between residents; therefore HCM development is the focus of the work described in this paper.

## Study population

The Welsh Demographic Service Dataset (WDSD) contains individual-level demographic and multiple address registrations for all individuals registered with a General Practice (GP) in Wales and is hosted anonymously in the SAIL Databank [28–31]. Data is held in two datasets: the individual-level dataset (WDSD-I) which contains: anonymous linking fields (ALF), sex, and week-of-birth and death; and the address-level dataset (WDSD-A) containing: RALF and address registration dates for each individual within a RALF.

For this study, individuals in WDSD-I were linked to WDSD-A using a common linking key where they were registered to a RALF at a specified index date). Where an individual had multiple RALF registration dates at index date, or was registered to multiple properties, the earliest registration record was retained. Records with a date-of-death prior to address registration end dates were removed. An index date of 1$^{st}$ January 2016 was chosen for illustration purposes, however, the methods can be run at any index date within the WDSD date range (early 1990's to current day).

## Household definition

A household was defined as all residents registered to the same RALF at index date, with household size calculated as the total number of residents.

Household composition measures were based on household membership, combined with age-based classifications to describe each household. Resident age was calculated at index date (1st January 2016) and used to define children as 17 years or under, and adults as 18 years or over. Households were further described as 'adult only', or 'family' depending on the absence or presence of a child aged resident. Households were grouped according to size and various household composition measures and presented as tabulations.

Sex and age were extracted from WDSD-I, with a derived tenancy duration measure created from WDSD-A for each resident (total number of days between the start of address registration and index date).

## Household composition measures

Survey methods obtain direct information from residents and allow classification of resident relationships. However, in the absence of direct relationship data, we used anonymised WDSD data, and therefore age-based assumptions were used for family generation classifications. Choice of age boundaries is difficult due to the natural variation between generations with literature showing variation in central tendency and spread [32–34]. We therefore made a pragmatic decision drawn from evidence of age ranges between child and parents [34]; analysis of generated data in this study describing age differences between residents as shown in S1 Fig could be used to inform future adaptations of the method to a data driven approach.

## Adult to child HCM

Household composition was defined by combinations of the total number of children and adults with possible outcomes: 'adult only', 'one adult with one or more children', 'two or more adults with children' (subdivided by one, two, or three or more children). A final category of 'children only' captures any household with no residents aged 18 or over. This method was designed to reflect similar measures used by Welsh Government [35, 36] based on UK Census data and projection methods to allow meaningful comparisons.

## Generational HCM: Relative age to youngest (AtY)

We allocated residents to one of three generations depending on the relative age difference to the youngest (AtY) resident. The youngest resident and any other residents with an age difference of 0–17 years were classified as generation-one (child or sibling), an age difference of 18–50 years as generation-two (parents), and an age difference of more than 50 years as generation-three (grandparents). Binary generation indicators were combined to create binary HCMs of *100* (one-generation), *110* (two-generation), *111* (three-generation), or *101* ('skip' generation); with count HCMs indicating the total number of residents in each category. For example, two residents aged 35 and 37 would be displayed as *100* (binary) and *200* (count); three residents aged 2, 4 and 30 as *110* (binary) and *210* (count); five residents aged 2, 3, 30, 45, 70 as *111* (binary) and *221* (count).

Refined HCMs were created, prefixed with 'family' or 'adult only' to differentiate households based on the age of the youngest resident as a proxy for household *maturity*. 'Family' households contained at least one resident classed as a child, and 'adult only' households contained no child aged residents.

**Table 1. Generalised household composition logic model.**

| HH Size | One-generation (Binary code: *100*) | Two-generation (Binary code: *110*) | Three-generation (Binary code: *111*) |
|---|---|---|---|
| 1 | Lone dwelling | | |
| 2 | Cohabiting partnerships | Single-parent families | |
| 3 | Larger adult groups/shared residences:<br>• house sharing professionals<br>• student accommodation<br>• care homes | Child & parent relationships:<br>• multiple combinations of residents including<br>• single-parent families<br>• two parents, 1 to 5 children<br>• other combinations and ability to split by age-based measures, e.g. 'adult only' or 'family' households | Multi-generations:<br>• child & parent & grandparent<br>• other combinations of non-family members |
| 4+ | | | |

Using these measures we aimed to establish a mechanism for classifying households according to a generalised household composition logic model as shown in Table 1.

## Generational HCM: Relative age to next (AtN)

Residents were allocated to a generational category depending on calculated age difference to the next eldest resident (AtN). The youngest, or index, resident was defined as generation one in all scenarios. Subsequently, the age difference to the next eldest resident was calculated and assessed; if the age difference was less than 18 years the individual was allocated to the same generation (no increment), an age difference of 18 years or more resulted in allocation to generation-two (generational indicator incremented by one). The process was repeated for all residents within the household. Binary, count, and age-based prefixed codes were created in the same approach as for the AtY method. With the AtN method, the two-generation 'skip' category was not a possible outcome as generations were incremented by a maximum of one generation. The AtY method limits outcomes to a maximum three generations, whereas the AtN method is restricted only by natural age expectancy, for example, a five-generation household could exist if five residents are aged at least 18 years apart.

Fig 1 describes a modelled example of two households illustrating how each resident contributed to the binary and count HCM classifications for AtY and AtN methods. The additional 'adult only' or 'family' prefix measures distinguish between similar household structures but different *household maturity*.

**Comparison of relative age methods.** To facilitate comparison of AtY and AtN methods, confusion matrices analysis was completed. Ideally, confusion matrix analysis requires a gold standard classification to test differences. However, in the absence of a gold standard method (using population-level linked data), we assumed AtY as gold standard and AtN as the test

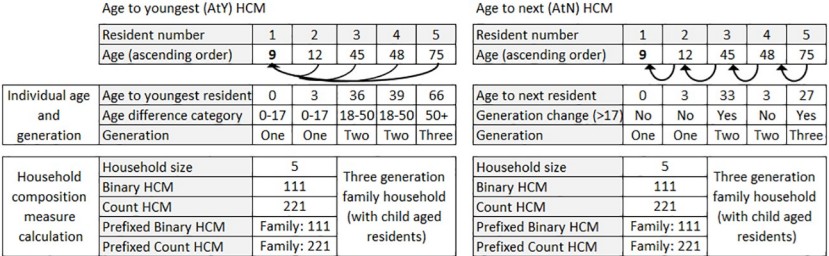

**Fig 1. Modelled examples of age to youngest (AtY), and age to next (AtN) HCM methods.**

model. To simplify comparison, results were restricted to households classed as one, two, or three-generation households in either method. To facilitate the comparison households classed as 'split' two-generation households from the AtY method were removed (N = 4,581 representing <0.5% of total properties), and any AtN household classification above three generations were reclassified as three generations (N = 606).

## Results

### Household and resident characteristics

The final linked dataset consisted of 3.09 million residents registered to 1.19 million properties in Wales at the index date of 1$^{st}$ January 2016. Records where an individuals' registered date of death preceded the address registration end date (n = 8,959) were removed along with duplicate records (n = 18). Results were restricted to households sized seven or less (N = 1,177,095) as they represent 99% of properties and 95% of residents. Fig 2 shows the total number of properties and population by households of up to seven with further detail on larger households shown in S2 Fig.

Table 2 displays property and population numbers by household size and shows that households sized four or less account for 91% of total properties, and 81% of the total population. Households are further detailed by sex, age, and tenancy duration with sub-grouping of households by 'family' or 'adult only' households (4,442 households with no adult aged residents were excluded, as well as child data from family household results to allow comparisons of adults). The proportion of females was generally higher in smaller households, most notably in 'family' households with 89% and 60% in households sized two, and three respectively. Age and household size were positively related, with 'adult only' households ranging from a mean of 61 years in single households to 38 years in households of seven. Age was generally uniform at 39 years in 'family' households, except for households of two residents being younger (36

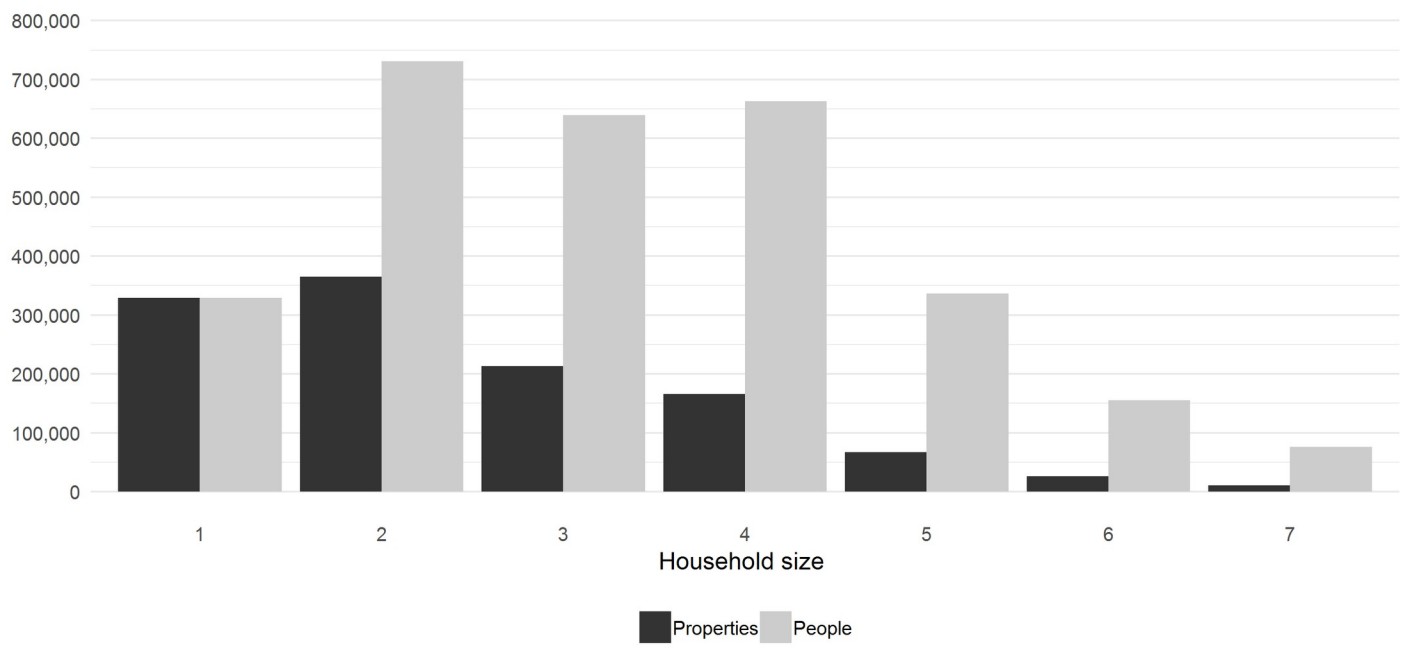

**Fig 2. Distribution of household size by properties and number of residents.**

**Table 2. Household metrics by household size for all, 'adult only' and 'family' households.**

| | Household Size | | | | | | |
|---|---|---|---|---|---|---|---|
| All households | 1 | 2 | 3 | 4 | 5 | 6 | 7 |
| Properties | 329,044 | 365,227 | 213,046 | 165,692 | 67,365 | 25,810 | 10,911 |
| Population | 329,044 | 730,454 | 639,138 | 662,768 | 336,825 | 154,860 | 76,377 |
| Female % | 55% | 52% | 49% | 49% | 48% | 48% | 48% |
| Age (mean(SD)) | 60.8(19.8) | 55.3(20.3) | 38.7(21.0) | 30.7(19.1) | 28.2(18.9) | 27.6(19.3) | 27.8(19.5) |
| Tenancy (mean(SD)) | 14.5(13.7) | 14.5(12.8) | 11.5(10.7) | 9.7(8.5) | 8.6(8.0) | 7.8(7.8) | 7.3(7.8) |
| | | | | | | | |
| Adult only households | 1 | 2 | 3 | 4 | 5 | 6 | 7 |
| Properties | 325,668 | 327,795 | 121,728 | 50,792 | 13,415 | 3,922 | 1,631 |
| Population | 325,668 | 655,590 | 365,184 | 203,168 | 67,075 | 23,532 | 11,417 |
| Female % | 56% | 50% | 45% | 44% | 43% | 42% | 42% |
| Age (mean(SD)) | 61.4(19.1) | 59.2(16.8) | 48.4(17.8) | 42.1(17.4) | 39.9(17.5) | 39.1(18.0) | 38.2(18.2) |
| Tenancy (mean(SD)) | 14.6(13.8) | 15.6(13.0) | 15.4(11.7) | 14.5(10.5) | 12.6(10.2) | 9.8(9.9) | 7.8(9.1) |
| | | | | | | | |
| Family households[a] | 1 | 2 | 3 | 4 | 5 | 6 | 7 |
| Properties | - | 36,660 | 91,115 | 114,836 | 53,932 | 21,881 | 9,278 |
| Population | - | 36,660 | 157,310 | 254,556 | 147,640 | 72,872 | 36,700 |
| Female % | - | 89% | 60% | 52% | 50% | 49% | 48% |
| Age (mean(SD)) | - | 36.1(10.7) | 38.9(11.1) | 39.3(11.3) | 38.8(13.1) | 39.0(14.3) | 39.3(14.8) |
| Tenancy (mean(SD)) | - | 5.3(6.0) | 7.2(7.0) | 8.8(7.2) | 9.2(8.1) | 9.2(8.6) | 9.0(8.8) |

[a] 'Family' households = households with at least one child and one adult; data displayed restricted to adults only

years). The relationship between household size and tenancy duration varied by 'adult only' and 'family' households. Adult households sized one to four ranged between 14 and 16 years of tenancy, decreasing to 8 years for seven resident households. Larger 'family' households of four or more have longer tenancy periods at around 9 years, with shorter tenancy noted in smaller households.

## Household composition

**Adult to child composition HCM.** Fig 3 shows results of the adult to child composition method. Adult only households account for the vast majority of households sized one and two, followed by a decreasing trend to 15% in the largest households of six and seven, with a corresponding increase of households with children. The proportion of single-parent households are at their largest in households of three residents at 12%.

For indicative purposes, comparisons were made with the Welsh Government (WG) 2016 published data [35, 36], which, as WG classification combines households of five or more residents, only households sized two to four are discussed here. For households sized two, household composition types were similar with a one percent variance with around 90% 'adult only' households and the remainder single-parent households. For households of size three, the proportion of single-parent households was equal at 12%, however, there was a disparity in 'adult only' households with 45% reported by WG compared to 57% in the SAIL method, and 42% (WG) compared to 31% (SAIL) for households with two adults and one child (varying methods should be noted, WG define dependent children as any person under 16 or under 18 and in full-time education [37]). A similar disparity was noted for households sized four, adult only households were reported as 20% by WG and 31% by SAIL, single-parent with children

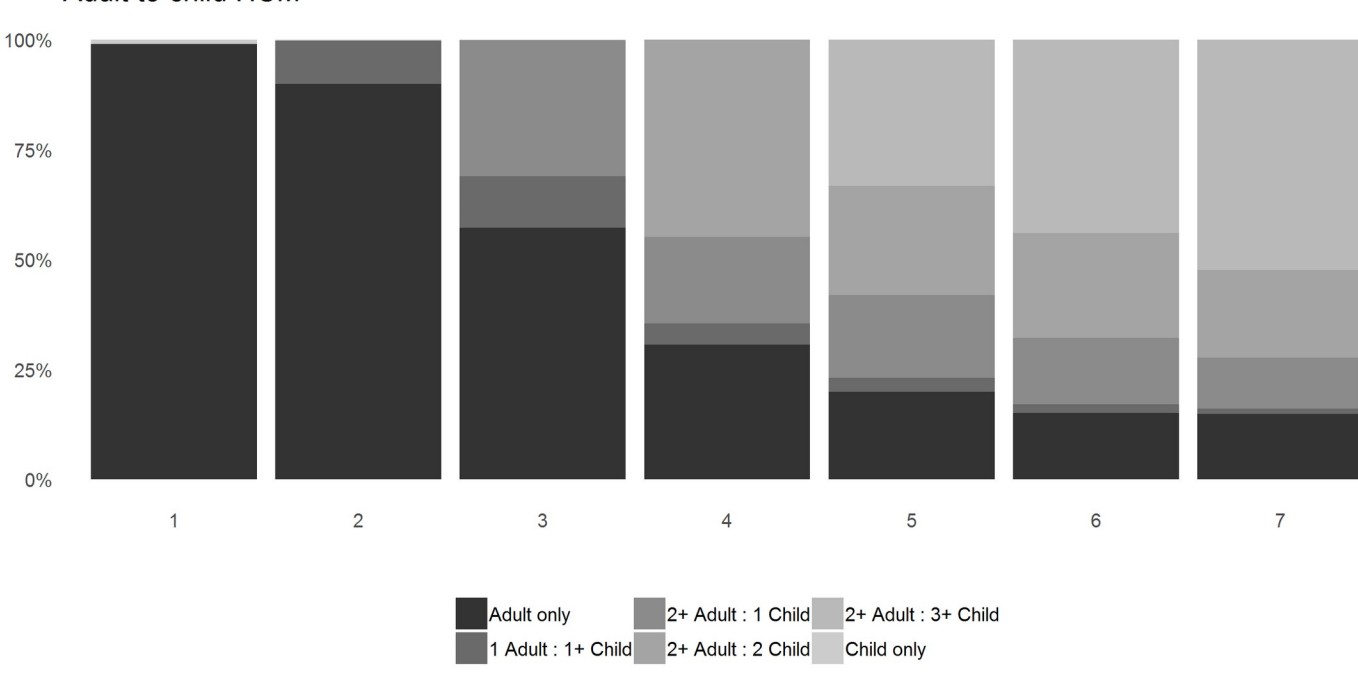

**Fig 3. Adult to child HCM.**

households were 6% for WG and 12% SAIL, with WG reporting 74% of households with two or more adults with one or more children compared to 65% for SAIL.

**Generational HCM binary methods: Age to youngest (AtY), and age to next (AtN).** Fig 4a and 4b show classification results for the binary AtY and AtN methods respectively. Around

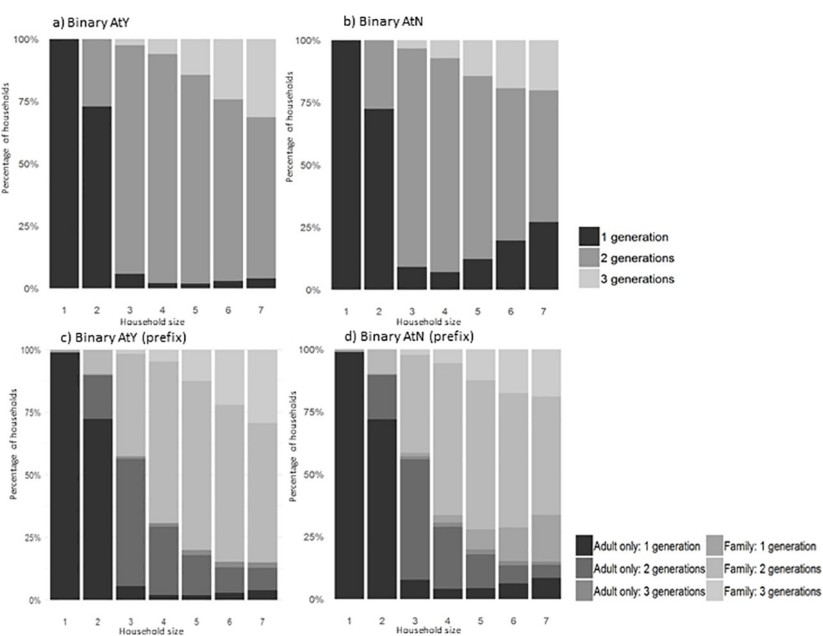

**Fig 4. Comparison of all households and using prefix split for AtY and AtN methods.**

75% of two resident households were classed as one-generation, the remainder by nature were two-generation households. The majority of larger households were two-generational, with an increasing proportion of three-generational households associated with increased household size, up to a maximum of 31% with the AtY method. The general pattern of results evident between methods is similar, with more variation in larger households with higher proportions of one-generational households in the AtN method compared to AtY.

The overall generational pattern is visible within the prefixed binary AtY and AtN methods in Fig 4c and 4d, but with additional differentiation between 'family' or 'adult only' type households. Commenting specifically on the AtY method, the notable feature is the division of two-generational households between the two household *types*. In three resident households, two-generational 'adult only' households account for half of all households and 'family' households account for 41%. This pattern reverses within increased household size; within households of four residents, 65% are two-generational 'family' households, and 27% are 'adult only', and households of seven have 56% 'family' households and 9% 'adult only' households. Household differentiation by type i.e. 'family' and 'adult only' is used for illustration purposes, however, more detailed stratification, for example using more granular age categories, can be achieved using SAIL data.

**Generational HCM count methods: Age to youngest (AtY), and age to next (AtN).** To this point, the generational measures presented are binary, indicating the presence or absence of at least one person within each generation. The following results indicate the total number, or count of residents per generation to provide further granularity into household composition. Fig 5a and 5b compare results for count HCM for AtY and AtN for 'adult only' households, with Fig 5c and 5d comparing results for 'family' households, sized two to seven.

Within 'family' households, the most common structure was a *nuclear family*, i.e. two parents and a corresponding number of children relative to household size; except for households of size two, which are predominantly single-parent families. In households of two to five,

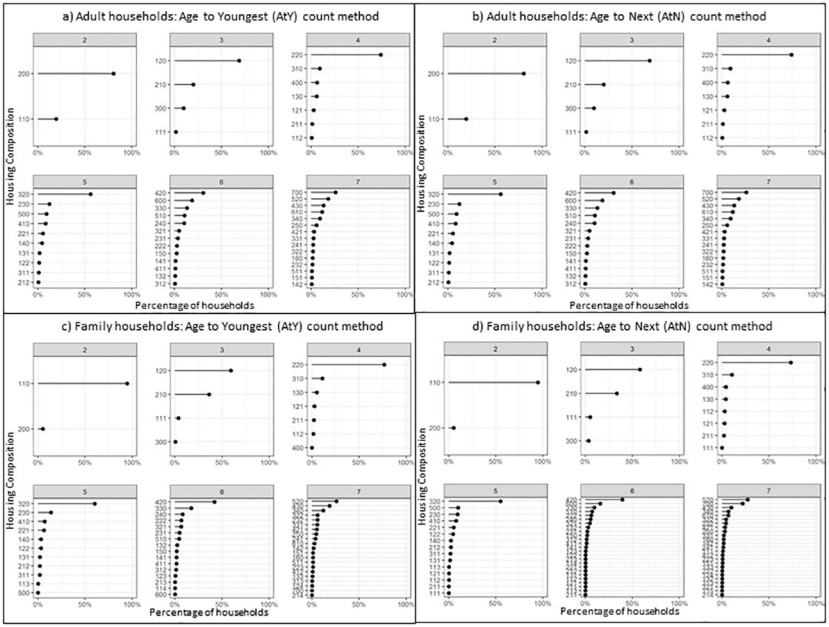

**Fig 5. Comparison of 'adult only' and 'family' households using count HCM AtY and AtN methods.**

the next most common structure indicates single-parent families. Larger households tend towards more than two adults in the parental generation category.

For 'adult only' households a similar pattern exists; showing a familial structure of more mature families with grown-up children, with two parents generally being the most common. Exceptions to this rule are noted in households of two indicating cohabiting partners, and households of size seven indicating larger groups of cohabiting adults of the same generation. Single generation households (for example, code: *700* for a seven-person single generation household) are more common compared to family structures in larger household sizes and suggest shared accommodation for larger groups of adults.

**Comparison of relative age HCM to Adult to Child composition methods.** Formal comparison of the *Adult to Child* HCM method compared to relative age HCM methods are not provided due to differences in approach, however, it is useful to comment on general differences between the type of results and insights provided. The *Adult to Child* HCM is more straightforward to interpret, and similar versions are currently used by government and statistical bodies in Wales [35]. However, compared to the AtY method, there is less insight into household composition. For example, households of size three using the A*dult to Child* HCM method reveals around 60% of households are 'adult only'; with AtY the 'adult only' group can be further split into 69% single child with two parents (code *120)*; 20% two children with single-parent (code *210)*; 10% three, same generation adults (code *300)*, with the remaining 1.5% being three-generation households (code *111*).

**Comparison of relative age HCM methods.** Visual differences between the two binary approaches can be seen in Fig 4a and 4b, with similar results and overall trends between methods. The main difference is that the AtN approach offers an increased number of one-generation households for larger households, compared to the results offered by AtY. As previously noted, the AtY method is used as a proxy for a gold standard classification method in the absence of a true gold standard to allow formal statistical-based comparisons through confusion matrices, the result of which show an accuracy of 96% for AtN compared to the *proxy* gold standard AtY. However, for larger households of five or more, the accuracy reduced to 87%, reinforcing what can be assumed from visual comparisons. The reason behind this separation is due to the varying methods of generational allocation. In a household with three residents aged 7, 22, and 35 the AtY method ignores the 22-year-old resident when assessing the age difference between the youngest and eldest residents; the eldest resident, being 28 years older, assumes classification as generation-two. However, with the AtN method, the middle resident 'bridges' the gap between youngest and eldest residents resulting in age to next resident of 15 and 13 years, which being less than 18 years results in all residents assigned to the same generation.

## Discussion

### Summary of main findings

We designed measures that can be applied to routinely-linked administrative data to group residents into households, to describe household composition, and benefit from RALF methodological enhancements. Two relative age-based algorithms were developed and returned similar results when applied to population and household-level data, describing household composition for 3.1 million individuals within 1.2 million households in Wales. To our knowledge, this is the first application of HCM at a population-level in Wales. Methods were designed flexibly to classify households at fixed or variable index dates to improve utility and enable use within various research settings and for varying study designs, including facilitating longitudinal research to model changes in living environments over time. Individuals'

anonymised address registration records are available in SAIL from the early 1990's to current day with new data added frequently; this adds flexibility to the methods which can be run at any specific index date, or at multiple time points to obtain measures of change for longitudinal studies.

As methods aimed to classify over three million residents they are by nature somewhat generalist; alterations to the methods for specific research projects could be expected to generate additional insight and accuracy. The methodology will be made available in the SAIL Databank and can be used as a basis for future methodological enhancements.

### Existing evidence

Physical and mental health, loneliness and isolation, caring needs, and child outcome research have all incorporated HCM to some degree to explore interactions and risk factors. A recent Public Health England report on COVID-19 outcomes noted that household composition was important to understand disparities but this measure was unavailable [38]; such studies could benefit from access to similar linked data and methods as we have described.

Research including HCM is predominantly based on surveys of relatively small numbers, or other estimated household measures [39]; our methods estimate household composition using individual-level data for a whole population. Current household composition metrics at government level are provided through detailed Census and survey results, an expensive and time consuming task, utilising the longitudinal data in the SAIL Databank we show HCM methods can describe household composition over a wide time period in an efficient manner.

### Implications for research, practice, and policy

Applications include enabling creation of richer datasets to better capture information and therefore understand more about individuals included in research studies. Linked data research is predominantly based on individual-level demographic, health, and social data, linked to area-level deprivation measures. Household-level data provides a conduit for additional data and insight into an individual's situation for use within population health research. Fig 6 describes the cyclical nature of how data relating to individuals, their households, and the wider community can flow and be harnessed to improve research. Individuals represented in Fig 6 would often be included in linked data research as disparate entities, however, HCM methods can group individuals and allow their separate situations to be described and applied to that group. For example, one child resident with serious ill-health could impact parents' wellbeing; incorporating this information could be valuable to understand health outcomes of those parents. Further examples include the household impact of living with a relative with dementia; or the increased household risk of COVID-19 contraction where a young resident tests positive for the disease.

A further application could be to supplement or improve linked administrative data. For instance, poorly recorded ethnicity data inhibits important research [40], and creation and application of a household ethnicity measure, pooling data from individuals in a household could, under certain considerations, be beneficial.

Longitudinal data in SAIL allows measurement and tracking of change in HCM and can assist with local and national government budgetary and public services planning, and could facilitate planning and measurement of interventions specifically targeted at tackling inequalities.

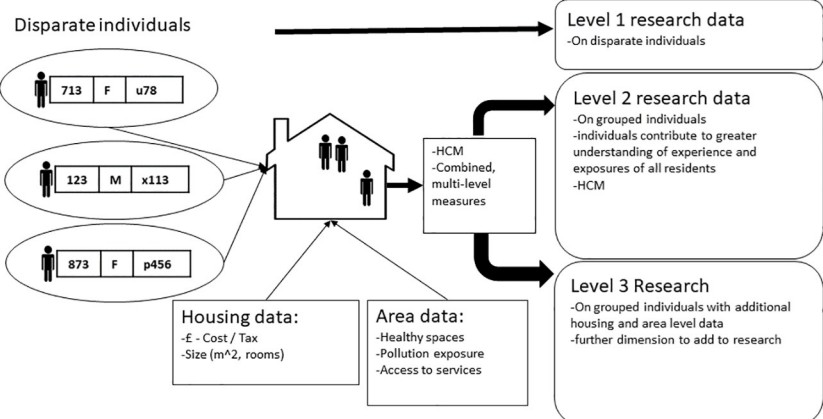

**Fig 6. Resident to household and community data model.**

## Strengths and limitations

The relative age methods described (AtY and AtN) provide further granularity, coverage, and insight compared to other currently available methods used for estimating household composition, such as in government reporting [35]. Whilst lacking finer detail associated with survey methods; these methods, using population-level linked data provide the ability to assign people to households, classify and describe households for research purposes, but importantly also allow for longitudinal studies. Methodological improvements to the RALF process provide a bedrock for future improvements in this area.

As a standalone measure, HCM can group residents within a household and describe HC for use in research studies, or understand change in HC over time for policy purposes. Topical examples for the application of HCM are evident in communicable disease transmission research, such as influenza and COVID-19 with outcomes linked to HC [2–6, 41]. Highlighting the relative strength of our methods, a recent study reported links between household size and positive COVID-19 tests (28), however, as noted by authors the HCM was limited to area-level estimates (based on 2011 postcode data). Our methods allow for individual- and household-level data so accuracy of household size is improved against such methods, as well as the availability of up to date longitudinal data.

The existing method could be used to infer certain dwelling types for specific research projects. For example, a household 'AtY' code of 7-0-0 in combination with average household age may allow student households to be selected for specific research aims.

Welsh Demographic Service address registration data is an important administrative data resource used primarily by the NHS for operational purposes but has some limitations for research. Accuracy of address registration records are dependent upon individuals updating GP's with address changes for themselves and family members, and accuracy is likely to vary between groups of people. Furthermore, there is no NHS 'point of capture' address verification necessitating subsequent data cleansing, and UPRN to address matching invariably causing some degree of error; problematic with blocks of flats or houses in multiple occupation, although implementation of the 'Parent: Child' UPRN system in SAIL has alleviated some error.

The HCM methods use two main assumptions which, as they are based on a theoretical approach to generational structure, should be understood. First, households are classified with an assumption that all household residents are related, and that generations consist of a child,

parent, or grandparent; further, fixed age differences informed generation boundaries. A general model is necessary to estimate complexities present in household composition in the real world and the assumptions used will introduce some estimate error.

## Future methodology improvements

Aforementioned considerations concentrate on internal household and individual-level data combinations, whereas future developments include linkage of administrative and Geographical Information System (GIS) derived data to refine housing type classifications enabling additional classification of houses in multiple occupation and communal residences (e.g. student housing and halls of residence). Additional data possibilities are numerous and include the ability to create measures such as access to health services, proximity to green space, air pollution exposures, and derivation of overcrowding measures using physical housing data.

## Conclusion

Improved RALF methods in SAIL have led to improved property level linkage, allowing design and application of household composition measures which assign individuals to shared residences and allow the description of household composition across Wales. HCM allow the selection of distinct household types which may be beneficial for research; it also enables data improvements in terms of quality and depth through mapping of individual characteristics and exposures to the household, and therefore all residents within. Such measures are important to help understand more detail about an individual's home and area environment and how that may affect the health and wellbeing of the individual, other residents, and potentially into the wider community.

## What is being added?

- We describe improvement to RALF methodology and potential future improvements;

- The first national-level description of household composition in Wales using population-level administrative data;

- We provide a population-level profile of household-level descriptive statistics in Wales;

- We provide various breakdowns of household composition through various HCM indicators;

- We provide the base for future work and improvements, and to allow wider inclusion of household-level data into numerous future research projects.

## Supporting information

**S1 Fig. Age distribution of individuals relative to youngest resident for AtY, and AtN methods.**
(TIF)

**S2 Fig. Household size distribution for households with over seven residents.**
(TIF)

## Acknowledgments

This study makes use of anonymised data held in the Secure Anonymised Information Linkage (SAIL) Databank. We would like to acknowledge all the data providers who make anonymised data available for research.

## Author Contributions

**Conceptualization:** Rhodri D. Johnson, Lucy J. Griffiths, Joe P. Hollinghurst, Ronan A. Lyons, Richard Fry.

**Data curation:** Rhodri D. Johnson, Ronan A. Lyons, Richard Fry.

**Formal analysis:** Rhodri D. Johnson.

**Funding acquisition:** Ronan A. Lyons.

**Investigation:** Rhodri D. Johnson.

**Methodology:** Rhodri D. Johnson, Lucy J. Griffiths, Ronan A. Lyons, Richard Fry.

**Project administration:** Rhodri D. Johnson.

**Resources:** Rhodri D. Johnson.

**Supervision:** Lucy J. Griffiths, Ronan A. Lyons, Richard Fry.

**Validation:** Rhodri D. Johnson, Lucy J. Griffiths, Joe P. Hollinghurst, Richard Fry.

**Visualization:** Rhodri D. Johnson.

**Writing – original draft:** Rhodri D. Johnson, Lucy J. Griffiths, Richard Fry.

**Writing – review & editing:** Rhodri D. Johnson, Lucy J. Griffiths, Joe P. Hollinghurst, Ashley Akbari, Alexandra Lee, Daniel A. Thompson, Ronan A. Lyons, Richard Fry.

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
