## [Decision Letter · Decision Letter 0]

1 Feb 2021

PONE-D-20-36605

Deriving household composition using population-scale Electronic Health Record data – a reproducible methodology

PLOS ONE

Dear Dr. Johnson,

Thank you for submitting your manuscript to PLOS ONE. After careful consideration, we feel that it has merit but does not fully meet PLOS ONE’s publication criteria as it currently stands. Therefore, we invite you to submit a revised version of the manuscript that addresses the points raised during the review process.

We look forward to receiving your revised manuscript.

Kind regards,

Sreeram V. Ramagopalan

Academic Editor

PLOS ONE

Journal Requirements:

2.) Thank you for stating the following financial disclosure:

'The funders had no role in study design, data collection and analysis, decision to publish, or preparation of the manuscript.'

3.) We note that you have indicated that data from this study are available upon request. PLOS only allows data to be available upon request if there are legal or ethical restrictions on sharing data publicly. For more information on unacceptable data access restrictions, please see http://journals.plos.org/plosone/s/data-availability#loc-unacceptable-data-access-restrictions.

Reviewers' comments:

Reviewer's Responses to Questions

**Comments to the Author**

1. Is the manuscript technically sound, and do the data support the conclusions?

Reviewer #1: Yes

2. Has the statistical analysis been performed appropriately and rigorously? 

Reviewer #1: Yes

3. Have the authors made all data underlying the findings in their manuscript fully available?

Reviewer #1: No

4. Is the manuscript presented in an intelligible fashion and written in standard English?

Reviewer #1: Yes

5. Review Comments to the Author

Reviewer #1: Is the manuscript technically sound, and do the data support the conclusions?

Yes, this is a comprehensive piece that clearly demonstrates a method for deriving household composition measures in the SAIL Databank, and linked population-level measures.

This is a well timed piece given the current pandemic, as household composition is a crucial and often missing piece of data. Indeed, beyond COVID these measures would have wide interest.

Has the statistical analysis been performed appropriately and rigorously?

Yes, this is a very well written, clear and comprehensive manuscript.

Have the authors made all data underlying the findings in their manuscript fully available?

No, not likely possible with these data.

Is the manuscript presented in an intelligible fashion and written in standard English?

Yes.

Additional comments

The flexible approach is a valuable aspect of the work. Perhaps it would be useful to expand a little more on this. My concern with the method was the ability to time update for household composition. If the census is completed every 10 years, how does time from census date affect the measures? Perhaps family households would change in the census interval faster than other household types.

Were any of the age differences data driven? Family dynamics are changing over time. Could this be accounted for in long-term studies?

Related, students and their living arrangements have been an important consideration for infection control efforts. Could the authors comment on whether the data is available (or might be available) to identify households with students?

Some more extensive footnotes for each of the figures would be useful so that they can be considered without referring back to the text.

6. PLOS authors have the option to publish the peer review history of their article (what does this mean?). If published, this will include your full peer review and any attached files.

Reviewer #1: No

---

## [Author Response · Author response to Decision Letter 0]

18 Feb 2021

Response to Reviewers:

We would like to thank the reviewers for their time and respond to their queries as follows:

“The flexible approach is a valuable aspect of the work. Perhaps it would be useful to expand a little more on this. My concern with the method was the ability to time update for household composition. If the census is completed every 10 years, how does time from census date affect the measures? Perhaps family households would change in the census interval faster than other household types.”

Thank you, we are pleased that the reviewer recognised that the flexible approach is a strength of the method. We have updated the article to emphasise the flexible approach whilst also taking the opportunity to remove ambiguity and to clarify the temporal nature of the data. Our term ‘census’ relates to the date at which we ran the algorithm for the date included in the manuscript. The method does not use ONS Census data but uses individuals’ GP address registration data in the Welsh Demographic Service data set (WDSD) held in anonymised form in the SAIL Databank. Our method can therefore be used to create the relevant household measures at any date across the date range available (1990 to almost current day – researchers can request the most up to date data which is within a month of being current https://data.ukserp.ac.uk/Asset/View/20). 

The query relating to how results may differ depending upon proximity to the census date are therefore not relevant within this context. However, as we are able to create results at any point this perhaps emphasises the flexibility of the approach and the ability to understand change over time. 

We have made the following manuscript revisions:

• Replaced the phrase ‘census date’ with ‘index date’ where appropriate to emphasise that the SAIL data used is not reliant upon ONS Census data

• Within the study population section, we added: “…however, the methods can be run at any index date within the WDSD date range (early 1990’s to current day).”

• Within the summary of main findings we added: “Individuals’ anonymised address registration records are available in SAIL from the early 1990’s to current day with new data added frequently; this adds flexibility to the methods which can be run at any specific index date, or at multiple time points to obtain measures of change for longitudinal studies”.

“Were any of the age differences data driven? Family dynamics are changing over time. Could this be accounted for in long-term studies?”

The age differences selected were not data driven but were based on a pragmatic approach of fixed age generational boundaries; we mention this limitation within the manuscript alongside the need for a general model. We also provide information relating to the age differences between residents in the supplementary file (S1) which could be used to inform a data driven approach, which could also be used to inform changes in the algorithm over time. 

Specific manuscript changes are as follows: 

• The following phrase has been added to the end of the household composition section: “…could be used to inform future adaptations of the method to a data driven approach”.

“Related, students and their living arrangements have been an important consideration for infection control efforts. Could the authors comment on whether the data is available (or might be available) to identify households with students?”

Thank you for raising this important issue. There is currently no specific indicator of student housing available in SAIL. To mitigate against the effects of capturing student residences the current study excluded larger households by limiting to those sized seven or less; therefore, larger student accommodation would not have been captured. However, a proxy measure could be used to infer whether a property was likely to be student accommodation. For instance, a household with a code of ‘7-0-0’ would indicate seven individuals 18 years of age of each other; combined with an indicator of age of youngest resident, or an average age, it could be inferred if the property was a student dwelling. This could further be developed using the LSOA to understand if the area was close to universities in Wales and increased chance of being a student dwelling. There would be difficulties in confidently differentiating between dwellings that may be shared residences but not students. Further work is currently underway with Welsh Government to link post 16 education data into SAIL to help further understand this population subgroup. 

Specific manuscript changes are as follows: 

• The following change has been made to the ‘Strengths and limitations’ section: “…The existing method could be used to infer certain dwelling types for specific research projects. For example, a household ‘AtY’ code of 7-0-0 in combination with average household age may allow student households to be selected for specific research aims”.

• The following rewording and additional change has been made to the ‘Future methodology improvements’ section: “…future developments include linkage of administrative and GIS derived data to refine housing type classifications enabling additional classification of houses in multiple occupation and communal residences (e.g. student housing and halls of residence)”. 

Some more extensive footnotes for each of the figures would be useful so that they can be considered without referring back to the text.

Unfortunately, PLOS ONE guidelines state that footnotes are not allowed and such information should be added to the main or reference text, therefore we have made no changes.

We have added a further reference [38] within the ‘Existing evidence’ section:

Harper G, Mayhew L. Using Administrative Data to Count and Classify Households with Local Applications. Appl Spat Anal Policy [Internet]. 2016 Dec 11 [cited 2021 Feb 8];9(4):433–62. Available from: http://research.dwp.gov.

---

## [Editor Report · Decision Letter 1]

22 Feb 2021

Deriving household composition using population-scale Electronic Health Record data – a reproducible methodology

PONE-D-20-36605R1

Dear Dr. Johnson,

We’re pleased to inform you that your manuscript has been judged scientifically suitable for publication and will be formally accepted for publication once it meets all outstanding technical requirements.

Kind regards,

Sreeram V. Ramagopalan

Academic Editor

PLOS ONE
---

## [Editor Report · Acceptance letter]

8 Mar 2021

PONE-D-20-36605R1 

Deriving household composition using population-scale Electronic Health Record data – a reproducible methodology 

Dear Dr. Johnson:

I'm pleased to inform you that your manuscript has been deemed suitable for publication in PLOS ONE. Congratulations! Your manuscript is now with our production department. 

Kind regards, 

on behalf of

Dr. Sreeram V. Ramagopalan 

Academic Editor

PLOS ONE